# CDK7 Predicts Worse Outcome in Head and Neck Squamous-Cell Cancer

**DOI:** 10.3390/cancers14030492

**Published:** 2022-01-19

**Authors:** Tobias Jagomast, Christian Idel, Luise Klapper, Patrick Kuppler, Anne Offermann, Eva Dreyer, Karl-Ludwig Bruchhage, Julika Ribbat-Idel, Sven Perner

**Affiliations:** 1Institute of Pathology, University of Luebeck and University Hospital Schleswig-Holstein, Campus Luebeck, Ratzeburger Allee 160, 23538 Luebeck, Germany; Luise.Klapper@student.uni-luebeck.de (L.K.); Patrick.Kuppler@uksh.de (P.K.); Anne.Offermann@uksh.de (A.O.); Eva.Dreyer@uksh.de (E.D.); Sven.Perner@uksh.de (S.P.); 2Department of Otorhinolaryngology, University of Luebeck, Ratzeburger Allee 160, 23538 Luebeck, Germany; Christian.Idel@uksh.de (C.I.); Karl-Ludwig.Bruchhage@uksh.de (K.-L.B.); 3Pathology, Research Center Borstel, Leibniz Lung Center, Parkallee 1-40, 23845 Borstel, Germany

**Keywords:** HNSCC, CDK7, pMED1, biomarker, TMA, prognosis, overall-survival

## Abstract

**Simple Summary:**

Head and neck squamous-cell cancer (HNSCC) is one of the most common cancer entities worldwide. Still, prognosis is rather poor, and therapeutic regimes come with harsh side-effects. In cancer research, certain biomolecules serve an important role as biomarkers that help to gain deeper insights into tumor biology. In this study, we sought to investigate the clinical implications of CDK7 and pMED1 as potential biomarkers in HNSCC patients. Therefore, we measured the levels of the above-mentioned proteins within tissue sections of our cohort using suitable software. Possible correlations of protein abundancy and clinicopathological features might help to guide future therapeutic decisions. We found that high levels of CDK7 in patients’ tumor tissue point to lower survival and disease-free survival rates. There are current studies that investigate CDK7 as a therapeutic target in drug therapy. Our study provides a rationale that such therapies might benefit HNSCC patients.

**Abstract:**

HNSCC is the sixth most common cancer worldwide and the prognosis is still poor. Here, we investigated the prognostic implications of CDK7 and pMED1. Both proteins affect transcription, and their expression is altered throughout different tumor entities. pMED1 is phosphorylated by CDK7. Importantly, CDK7 and MED1 have been ascribed prognostic implications by various studies. However, their prognostic value in head and neck squamous-cell cancer (HNSCC) remains elusive. We applied immunohistochemical staining of CDK7 and pMED1 on our large and clinically well-characterized HNSCC tissue cohort comprising 419 patients. Software-aided quantification of staining intensity was performed as a measure of protein expression. The following results were linked to the clinicopathological features of our cohort and correlated in different tissue types (primary tumor, lymph node metastasis, distant metastasis, recurrence). Upregulation CDK7 was associated with worse 5-year overall survival as well as disease-free survival in HNSCC while being independent of other known prognostic factors such as p16-status. Also, CDK7 expression was significantly elevated in immune cell infiltrated tumors. In HNSCC CDK7 might serve as a novel prognostic marker to indicate the prognosis of patients. Furthermore, in vitro studies proved the feasibility of CDK7 inhibition with attenuating effects on cell proliferation underlining its remarkable translational potential for future therapeutic regimes.

## 1. Introduction

Head and neck squamous-cell cancer (HNSCC) comprises a group of carcinomas that arise from the mucosal lining of the larynx, pharynx, and oral cavity [1,2]. Worldwide, HNSCC is the sixth most common cancer entity [3,4]. Prognosis is still rather poor. HNSCC carcinomas are stratified by UICC classification. Whereby stages I/II are considered as early stages and stages III/IV as advanced stages. Most patients are diagnosed at advanced stages when the prognosis is significantly worse, and recurrences are more frequently observed [5].

Despite considerable achievements in cancer treatment in general, therapy for HNSCC has not changed much over the past decades. Surgery, radiotherapy, and chemotherapy remain the cornerstones in most regimes [6]. Although various other systemic and topical treatments have been proposed, none seem to improve the patient’s prognosis [7]. While early stages are treated with surgery alone, advanced stages require a multi-modal approach [1], often accompanied by serious side effects as patients suffer from functional impairments in the surgical field and/or necrosis, fibrosis, and dysphagia caused by chemoradiotherapy [8].

As yet, there are no reliable biomarkers that justify therapy de-intensification. In this respect, p16 has been investigated in the De-ESCALaTE trial and proved insufficient [9]. p16 is a well-known biomarker expressed in HPV-associated carcinomas that commonly have a better prognosis [1,10,11]. In addition, hypermethylation of cyclin-dependent kinase inhibitor 2A (CDKN2a), a gene coding for tumor-suppressor genes p14^ARF^ and p16^INK4a^, is involved in tumorigenesis of HNSCC and has also been investigated as a prognostic marker for this entity. Methylation transcriptionally silences genes and consequently decreases protein levels. However, investigation results varied between the studies regarding prognostic implications of CDKN2a [12]. Tumor-suppressor gene p53 is one of the most commonly altered genes in HNSCC. Tumors harboring high-risk mutations showed worse outcomes [13]. Nonetheless, an IHC study carried out to evaluate the association of p53 protein expression to clinicopathological features of an HNSCC cohort could not prove prognostic implications [14].

One obstacle is the heterogeneous nature of HNSCC that comprises carcinomas from different locations with varying biomolecular traits and mutational patterns that hinder the exploration of novel biomarkers as well as pharmaceutical inhibitable oncogenic key drivers [6]. Consequently, ubiquitously expressed proteins that sustain malignant processes in cancer cells might be favorable targets for future research [15].

Defects of cell-cycle control and transcription are often present in cancer and transcriptional addiction is a well-known trait of cancer cells. Transcriptional addiction describes the reliance of cancer cells on specific transcriptional programs [16]. Disturbance of these programs is lethal for cancer cells. Transcriptional addiction is often established through so-called super-enhancer sites, an accumulation of enhancers that strongly drive transcription of their dependent genes [16,17].

Cyclin dependent Kinase 7 (CDK7) is a key regulator of transcription and cell-cycle control. In cell-cycle control, CDK7 is the only known cyclin-activating kinase (CAK) and therefore coordinates the interplay of other cyclin-dependent kinases in cell-cycle progression [18]. In transcription, CDK7 is part of transcription factor II H and enables proper loading of DRB sensitivity-inducing factor (DSIF) and negative elongation factor (NELF) by phosphorylation of the C-terminal domain of RNA polymerase II (RNA Pol II) which in turn facilitates RNA Pol II pausing. This step is mandatory for effective transcription and the integration of activating signals [19,20]. Recent investigations have shown that CDK7 can drive transcriptional addiction [21,22,23] and it also affects genomic stability [24]. Besides inhibition, CDK7 has shown promising results in preclinical models of diverse entities reflecting its great translational potential [15,24,25,26,27,28].

A protein that contributes to CDK7 impact on transcription is mediator of RNA polymerase II transcription subunit 1 (MED1) as part of the mediator complex. MED1 is phosphorylated by CDK7 at residue T1457 and then referred to as pMED1. Phosphorylation reinforces its association with the mediator complex [29]. Functionally pMED1 is a global coactivator of transcription. In particular, alterations in the expression of the MED1 subunit have been repeatedly reported in the context of metastatic disease throughout different cancer entities [30,31,32].

Deregulation of CDK7 and MED1 in cancer has been observed in many studies and linked to aggressive clinicopathological features and worse prognosis [10,30,31,32,33,34,35,36]. Based on their biological properties we hypothesized that these proteins might possess promising prognostic value in HNSCC. However, their prognostic role and potential interaction in HNSCC have not yet been studied. Therefore, we investigated the expression of CDK7 and pMED1 in HNSCC using immunohistochemical staining (IHC). Staining was applied and digitally quantified on a large tissue microarray (TMA) HNSCC cohort. The retrieved results were correlated with clinicopathological features.

## 2. Materials and Methods

### 2.1. Tumor Material

Our study was approved by the Ethics Committee of the University of Luebeck (project code AZ 16-277, date of approval 18 November 2016) and conducted in accordance with the Declaration of Helsinki. We used our HNSCC cohort for the analysis of CDK7 and pMED1 expression. Previously, our group set up a cohort of 419 HNSCC patients [37,38,39,40,41]. All patients were diagnosed between 2012 and 2015 in our Institute of Pathology and treated in the Department for Otorhinolaryngology of the University Medical Center Schleswig-Holstein in Luebeck. Clinical data were obtained from medical records. Our cohort comprised 22.5% female patients and 77.5% male; 87.6% of all patients consumed nicotine (at least one pack per year) and 43.1% consumed alcohol. The primary tumor site was broken down as follows: 47.3% of all tumors were in located the pharynx (12.6% hypopharyngeal and 34.7% oropharyngeal), 28.1% were laryngeal HNSCCs, 21.4% originated from the oral cavity, and the remaining 3.2% were identified as cancers of unknown primary. Regarding p16 status, 28.9% of all patients were p16-positive. In 25.1% of the cases, recurrences were detected during follow-up. At initial diagnosis tumor stages were determined: 52.5% of the patients had tumor stages T1/2 and 47.5% T3/4; lymph node metastasis was present in 57.2% of the cases while distant metastasis was present in 13.4%. In addition, 41.2% of tumors were classified as UICC I/II and 58.8% as UICC III/IV, 76.2% had histological grading of G 1/2, the rest was assigned grading of G 3/4. TNM stages were assigned according to the 8th edition of the TNM classification for HNSCC.

For our analysis of CDK7 and pMED1 expression, we used tissue of primary tumors (PTs) (*n* = 326), lymph node metastasis (LNs) (*n* = 162), distant metastasis (DMs) (*n* = 17), and recurrences (RDs) (*n* = 77). Formalin-fixed paraffin-embedded (FFPE) tumor tissues were collected, and TMAs were prepared, as previously described [37,42]. Each TMA comprised at least one triplet of cores per patient. A single core was 1mm^2^ in size. Up to 54 tumor samples and six samples of benign tissue from the head and neck region were arranged per TMA slide.

### 2.2. Immunohistochemistry

Immunohistochemical staining was performed on 4 μm thin sections of FFPE tissue. Sections were deparaffinized and antigens were heat-mediated retrieved, as previously described [43]. Automated staining was applied using the automated Ventana BenchMark staining system and the IViewDAB Detection Kit (both Roche, Basel, Switzerland). For the staining of CDK7 and pMED1, the following antibodies were used at the indicated dilution after successful control tissue staining according to data sheets: CDK7 (mouse monoclonal, CDK7 (MO1) Mouse mAb, 1:500, Cell Signaling, Danvers, MA, USA) and pMED1 (rabbit polyclonal, Anti-TRAP220/MED1 (phospho T1457) antibody, 1:100, Abcam, Cambridge, UK). Slides were then counterstained with hematoxylin and bluing reagent.

### 2.3. Digitalization and Evaluation

Using a Ventana iScan HT scanner (Ventana, Tucson, AZ, USA) the slides were digitalized, and “.bif” files were generated. For evaluation of the scanned slides, the semi-automatic commercially available image analysis software Definiens Tissue Studio^®^ (Definiens Developer XD 2.0, Definiens AG, Munich, Germany) was employed. This software allows objective assessment of the staining intensity in different cellular compartments within specified regions of interest (ROI). ROIs were defined as tumor cell areas that were manually annotated in each TMA core to exclude stromal cells and benign areas from evaluation. In addition, cores that comprised staining artifacts or tissue folds were excluded from later analysis. Staining intensity was measured on a continuous arbitrary scale to reflect protein expression of the cells. Data acquisition was performed on a Windows-7-based computer with a 24″ monitor and resolution 1920 × 1080 px for Definiens Tissue Studio^®^.

### 2.4. Statistics and Visualization

For statistical analysis, the mean of the patients’ triplets was obtained for each measurement. For the statistical analyses and data visualization, R software (version 4.0.2, R Foundation, Vienna, Austria; http://www.R-project.org, accessed on 8 December 2021) was used. For correlation of CDK7 and pMED1 measurements, the Pearson correlation coefficient (PCC) was calculated. ANOVA was performed to compare CDK7 expression at different primary tumor locations and between different tissue types. Unpaired two-tailed t-test was applied to discriminate alterations in expression concerning clinicopathological parameters of the patients. In cases where multiple testing was performed *p*-values were adjusted by the Benjamini–Hochberg procedure. Five-year overall survival (OS) and disease-free survival (DFS) are illustrated as Kaplan–Meier curves and were compared using the log-rank test. *p*-values less than 0.05 were considered statistically significant. Boxplots show the median and the interquartile range as error bars.

## 3. Results

### 3.1. Distinct Staining

CDK7 and pMED1 were both stained within the nuclei. Expression patterns for CDK7 and pMED1 in our HNSCC cohort were homogenous. The intratumoral difference in expression of the cells was negligible. Overall expression varied between patients. We observed a range of expression intensities, while some triplets showed no expression others displayed strong immunoreactivity (see Figure 1). To adequately address these patterns the mean staining intensity, present in the nuclei was obtained for each core and considered as a measure of protein expression. The likewise-obtained expression levels were used for further analysis.

### 3.2. CDK7 Overexpression Is Associated with a Shorter Overall and Disease-Free Survival

Kaplan–Meier plots for OS and DFS were created, and log-rank testing was applied. For the following analysis, our cohort was dichotomized by median CDK7 expression, thereby creating two groups (high expression above median: *n* = 160, low expression below median: *n* = 158) (see Table 1). Patients with high CDK7 expression had a significantly shorter OS as well as DFS (OS: *p* = 0.037, DFS: *p* = 0.016). For OS, the 5-year survival rates were estimated at 54% for CDK7 high expression and 66% for low expression, while disease-free survival rates were 42% and 58% for high and low CDK7 expression, respectively (see Figure 2). Subsequently, univariable and multivariable Cox regression was performed to confirm if the prognostic value of CDK7 expression was independent of other prognostic factors for 5-year OS and DFS. To adjust for these factors, univariable Cox regression was first calculated followed by multivariable Cox regression in consideration of possible interactions. High CDK7 expression was an independent prognostic factor for both OS and DFS (OS: HR = 1.48 (95% CI 1.02–2.15), *p* = 0.038, DFS: HR = 1.51 (95% CI 1.08–2.10), *p* = 0.016). In addition, alcohol and packyears, p16 status, T-stage, M-stage, and UICC stage were identified as prognostic factors for 5-year OS in our cohort. For DFS, age was an additional prognostic factor. In multivariable Cox regression, high CDK7 expression remained significant and therefore predicted prognosis for OS and DFS independently of factors listed above (OS: HR = 1.50 (95% CI 1.01–2.22), *p* = 0.045, DFS: HR = 1.50 (95% CI 1.05–2.14), *p* = 0.024). The results of our analysis are summarized in Table 2.

We next assessed differences in CDK7 expression concerning clinicopathological parameters other than disease-free and overall survival. Our cohort was subdivided into patients older or younger than the median age (62 years). The difference in expressions of CDK7 within the PTs of these groups was not significant (*p* = 0.17). We also correlated other dichotomous clinical features with CDK7 expression in the PTs. However, no significant results could be obtained. Table 3 summarizes our findings.

### 3.3. Congruent CDK7 Expression in Different Primary Tumor Locations

CDK7 expression was examined in different sites of HNSCC PTs. However, comparing these tumor sites showed no significance (ANOVA = 0.81). The highest mean expression was detected in oropharynx carcinomas p16+ HNSCC (*n* = 55), while the lowest expression was observed in oral cavity carcinomas (*n* = 74) (see Figure 3).

### 3.4. Lower CDK7 Expression in Lymph-Node Metastasis Than in Primary Tumors

We compared CDK7 expression in different tumor tissues of the PTs, LNs, DMs, and RDs to verify if expression changes occur during tumor progression. ANOVA testing revealed significant differences between the tissue types (*p* = 0.00032), post-hoc *t*-test showed expression in LNs (*n* = 162) to be significantly lower than in PTs (*n* = 326) (*p* < 0.0001, adjusted) (see Figure 3).

### 3.5. CDK7 Levels Are Higher in Immune-Infiltrated Tumors

We determined the immune cell infiltrate patterns of tumors within our cohort as previously described [40]. Briefly, we distinguished three different tumor immune profiles: hot tumors were declared tumors with diffuse immune cell infiltration (*n* = 65), excluded tumors were those with immune cells predominantly present in the tumor stroma (*n* = 140), and cold tumors contained only small numbers of immune cells (*n* = 67). We hypothesized that hot tumors especially might exhibit a low level of CDK7 as Zhang et al. could show that this occurrence might be associated with increased immune cell influx [24]. Interestingly, we found that hot tumors comprised significantly higher levels of CDK7 than cold and excluded tumors (hot vs. cold, *p* = 0.045, hot vs. excluded, *p* = 0.044) (see Figure 4). We then matched CDK7 expression with PD-L1 TPS and observed significantly higher TPS in CDK7-overexpressing tumors (*p* = 0.0063).

### 3.6. Correlating CDK7 and pMED1 Expression

To investigate a possible role of CDK7-mediated MED1 phosphorylation we correlated CDK7 and pMED1 expression in tissue of PTs (*n* = 320), LNs (*n* = 157), RDs (*n* = 70), and DMs (*n* = 14). We observed a significant correlation within all tumor sites (DMs: *p* = 0.0019, the rest: *p* < 0.001). PCC ranged between 0.39 to 0.71, with the lowest correlation detected in LNs (PCC = 0.39) and the highest correlation in DMs (PCC = 0.71). In PTs, PCC was calculated at 0.42 and in RDs, 0.59 (see Figure 5).

### 3.7. Prognostic Implications of CDK7 and pMED1 Co-Expression

Under our working hypothesis of CDK7-mediated pMED1 phosphorylation, we further explored prognostic relevance based on the combined expression of these two proteins. As mentioned in the introduction high CDK7 expression might lead to increased phosphorylation of MED1 and consequently to stimulation of gene transcription including among oncogenes.

For this analysis survival data of 314 patients was applicable. The cohort was divided based on the medians of CDK7 and pMED1 expression in the PTs, yielding a total of four groups as follows: one group with both proteins highly expressed (*n* = 103), one group with both proteins slightly expressed (*n* = 102), and two groups, each with only one of the proteins strongly expressed (only CDK7 highly expressed: *n* = 55, only pMED1 highly expressed: *n* = 54). No significant differences in 5-year overall survival (OS) or 5-year disease-free survival (DFS) were revealed in the log-rank test (*p* = 0.11 and *p* = 0.067, respectively). The lowest OS and DFS rates were observed for the CDK7 high- and pMED1 low-expressing group with a 46% survival rate and 38% disease-free survival rate, respectively. The CDK7 low and pMED1 low group revealed the highest survival rates of 69% for 5-year OS and 63% for 5-year DFS (see Figure 6).

Interestingly, subgroups assembled according to CDK7 expression. Generally, high CDK7 expression hinted at a shorter 5-year OS and DFS, irrespective of pMED1 expression, while low CDK7 expression indicated longer 5-year OS and DFS. We concluded CDK7 by itself might have stronger prognostic implications than combined CDK7 pMED1 status.

## 4. Discussion

To the best of our knowledge, this is the first study to investigate the potential role of CDK7 and pMED1 in HNSCC. We could show that the expression of CDK7 and pMED1 are correlated throughout different tumor sites. This might hint at a potential functional linkage of these two proteins. We did not observe significant differences in CDK7 expression by tumor site. However, our data revealed a prognostic drawback for patients with high CDK7 expression. Considering the current lack of appropriate biomarkers for risk stratification and harmful therapeutic approaches, CDK7 might be suggested as a valuable new biomarker. Importantly, survival analysis in patients with CDK7 overexpression showed significantly worse 5-year OS as well as DFS rates when compared to patients with lower CDK7 levels. Including CDK7 in the routine pathological workup might add meaningful information about patients’ prognoses as well as guide follow-up management.

Few studies have addressed the interaction of CDK7 and MED1. Most commonly, pMED1 is recruited to the mediator complex upon nuclear receptor (NR) signaling. The specific role of pMED1 and its participation in oncogenic processes remain elusive as pMED1 seems to have varying prognostic implications. Regarding entities where NR signaling is a well-known trait, pMED1 most likely acts in an oncogenic manner and thus is linked to worse outcomes in breast cancer and prostate cancer [26,44]. On the contrary, in NR-unrelated tumors, pMED1 levels often decrease during progression, as described for bladder cancer [36] and various other entities [30,31,32]. However, for our HNSCC cohort, we did not observe significant effects on survival based on pMED1 expression. As evidence is lacking as to whether NR-signaling serves as a key driver in HNSCC progression [45], we speculate that CDK7-dependent phosphorylation of MED1 is of minor importance for the transcriptional effects mediated by CDK7 in HNSCC. Thus, CDK7 overexpression might elicit its effect through the general activation of transcription in HNSCC. In this respect, super-enhancer (SE)-driven transcription is especially reliant on CDK7. A large number of transcriptional proteins bind to SEs resulting in increased transcription of dependent genes which are predominantly part of cell identity and survival [16]. In addition, in HNSCC SEs play an important role. Finally, investigations have shown that SEs are crucial for the survival of HNSCC cancer stem cells. Perturbation of SE-mediated transcription led to the elimination of these cells [46].

CDK7 overexpression has been observed in many cancer entities and generally indicates a worse prognosis and is associated with aggressive clinicopathological features. In oral squamous-cell cancer, CDK7 overexpression was associated with shortened OS and DFS, as well as being linked to a higher T-stage [35]. Similarly, studies conducted on gastric cancer, esophageal squamous cell carcinoma, and ovarian cancer showed worse survival, higher grading, and advanced tumor stages related to increased CDK7 levels [10,33,34].

Inhibition of CDK7 has been thoroughly investigated in cell culture and mice models with promising results. It has long been thought that inhibition of transcription leads to global impairment of cell viability. With increasing knowledge about transcriptional addiction, however, it became clear that cancer cells are more sensitive to the interference of transcriptional programs than benign cells [16,17]. Therefore, inhibition might be employed in future therapeutic regimes. In HNSCC, administration of THZ1 (a CDK7 inhibitor) caused impairment of cell proliferation and induced apoptosis in vitro [23,27]. In SCLC, depletion of CDK7 led to genomic instability, triggering increased immune infiltration that had synergistic effects with simultaneous anti-checkpoint therapy. This is of great interest as it raises the immune checkpoint therapy response [24]. We suppose that a combinational treatment might be of advantage for HNSCC patients as well, as immunotherapy develops as an integral component of therapeutic regimes. In particular, patients with high CDK7 expression might benefit since CDK7 was recently discovered to increase PD-L1 expression through an MYC-dependent pathway in NSCLC. This mechanism leads to immune escape in the presence of infiltrating immune cells. The authors suggest that CDK7 inhibition sensitizes tumor cells to therapy with immune checkpoint inhibitors [47]. We also could observe higher PD-L1 TPS in CDK7-overexpressing tumors, concluding that similar pathways might facilitate immune escape in HNSCC as well, and further supporting the statement that particularly patients with high CDK7 expression might profit from immunotherapy. In addition, we sought to investigate the association of CDK7 expression and immune infiltration for our cohort, as low CDK7, as mentioned above, might elicit immune infiltration. However, we found significantly higher CDK7 expression in immune-infiltrated tumor tissue. Previously we described higher TPS in immune-infiltrated tumors [48]. We suppose that CDK7 might drive higher PD-L1 expression in these tumors. However, since our study is the first to describe this linkage, further research is needed to characterize the tumor microenvironment according to CDK7 expression in HNSCC. We appreciate the limitations of our study as it provides correlation of clinical data and CDK7 expression determined via IHC staining. Further in-vitro studies are needed to investigate causation.

However, there is a major translational potential as CDK7 does not only have prognostic ramifications but can also be effectively inhibited with anti-neoplastic effects. As we provide evidence that low CDK7 expression is linked with a better prognosis one might expect that drugs targeting CDK7 in HNSCC could be beneficial for their anti-oncogenic function.

## 5. Conclusions

Our data suggest that CDK7 harbors valuable prognostic implications for HNSCC. Considering the poor prognosis and sparse landscape of suitable biomarkers and druggable targets in HNSCC it might be worth further pursuing CDK7 as a potential protein with implications for both prognosis and therapy, as there are already functioning inhibitors under investigation in the early stages of clinical trials [22,49]. In particular, the linkage of CDK7 and immunotherapy might be of great interest for prospective research since checkpoint inhibition is becoming increasingly important in HNSCC treatment but mechanisms to boost therapy response remain poorly understood. Future studies may gather additional justification to include CDK7 in the routine pathological workup and to potentially carry out clinical trials to assess inhibiting agents in HNSCC. Despite phosphorylation of MED1 by CDK7, their interactions seem to play a secondary role in the clinical context of HNSCC. Finally, prospective studies are necessary to validate our findings on independent cohorts and to unravel molecular mechanisms by which CDK7 contributes to poor prognosis in HNSCC.

## Figures and Tables

**Figure 1 cancers-14-00492-f001:**
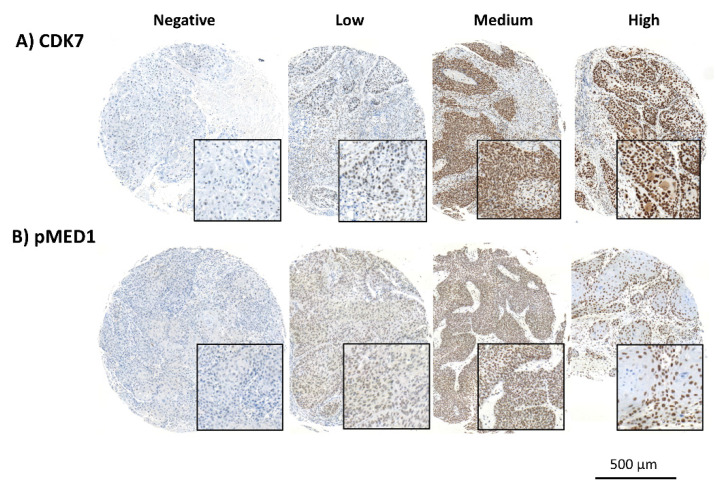
Expression patterns for (**A**) CDK7 and (**B**) pMED1 in our HNSCC cohort (magnification 20×, insert 30×). TMAs were immunohistochemically stained for CDK7 and pMED1. The cancer specimen showed a variation in protein expression. The expression of the proteins was homogenous within the patients’ samples, meaning there was little disparity within the staining of cells in a single core. In some HNSCCs the cancer cells showed no staining and accordingly protein expression was considered negative. Throughout the cohort a range of staining intensities was obtained, (**A**,**B**) illustrate samples of a negative, low, medium, and high protein-expressing tumor specimens.

**Figure 2 cancers-14-00492-f002:**
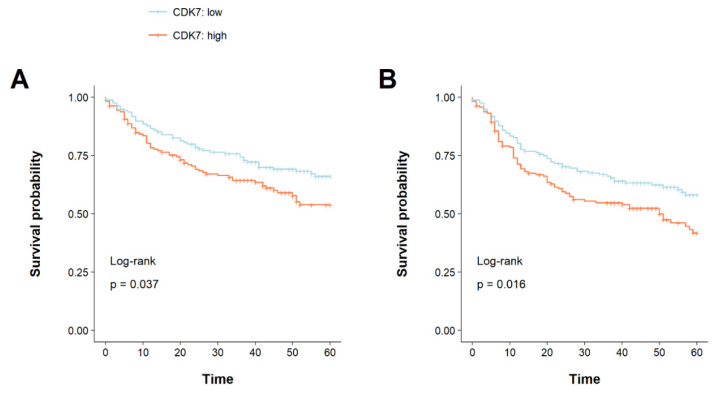
Kaplan–Meier graphs with a *p*-value of log-rank test of (**A**) 5-year overall survival and (**B**) 5-year disease-free survival. The cohort was stratified into two groups based on the median expression of CDK7. Expression above the median was considered as high expression conversely expression below the median was considered as low expression. Upregulation of CDK7 correlated with a significantly shorter OS as well as DFS (*p* = 0.037, *p* = 0.016).

**Figure 3 cancers-14-00492-f003:**
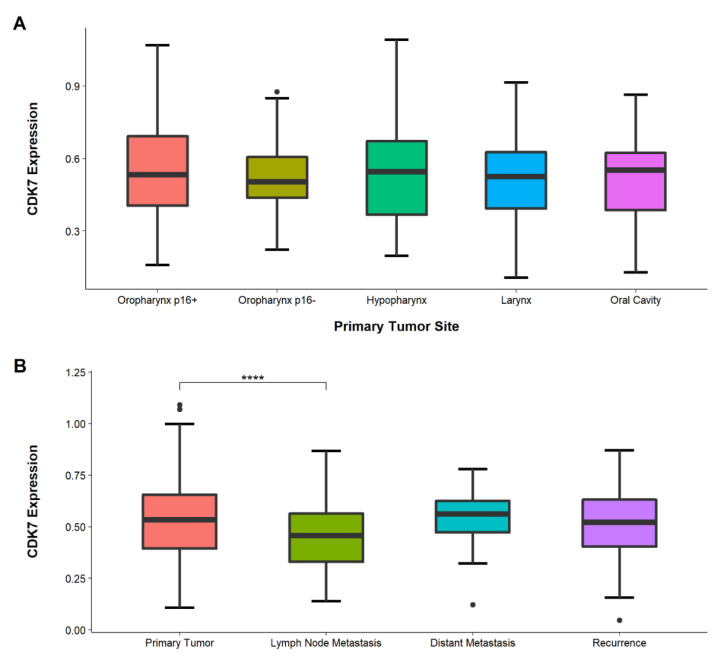
(**A**) Boxplot showing the expression of CDK7 in different HNSCC tumor sites. No significant difference was observed between the locations. (**B**) Boxplot comparing the expression of CDK7 in PTs, LNs, DMs, and RDs. CDK7 expression in LNs was significantly lower compared to PTs (*p* < 0.0001, adjusted). Other comparisons did not reach significance (*p* > 0.05). **** *p* < 0.0001, ● Outliers.

**Figure 4 cancers-14-00492-f004:**
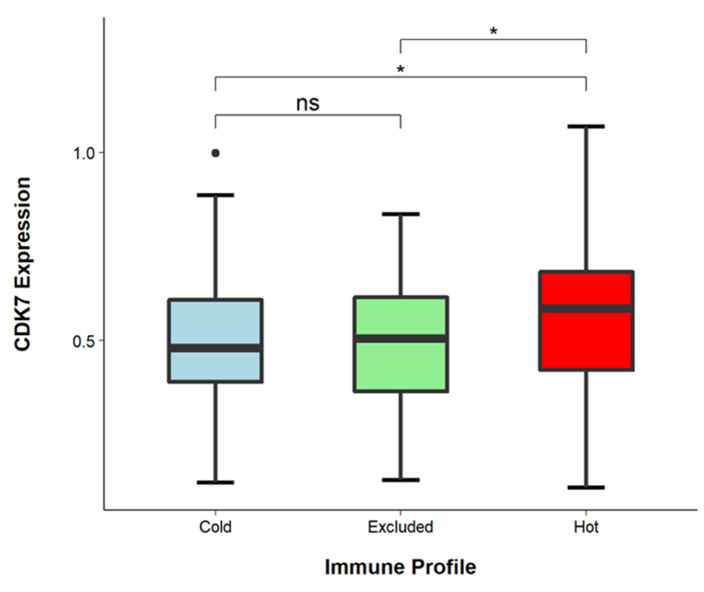
CDK7 expression in different tumor immune profiles. Cold tumors had few to no immune cells; in excluded tumors, immune cells were only present in the tumor stroma; and in hot tumors, immune cells were diffusely infiltrating in between the tumor cells. Compared to cold and excluded tumors, hot tumors harbored a significantly higher CDK7 expression (hot vs. cold *p* = 0.045, hot vs. excluded *p* = 0.044), while cold and excluded tumors showed no significant difference (*p* = 0.91). * *p* < 0.05, ● Outliers.

**Figure 5 cancers-14-00492-f005:**
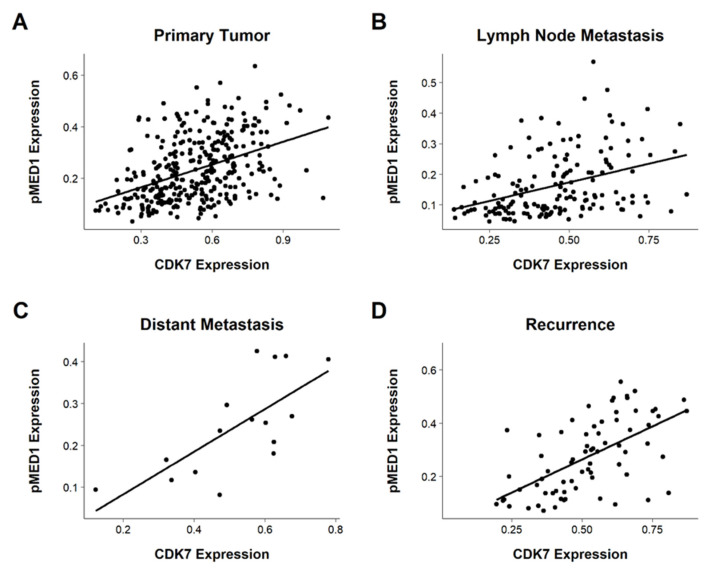
Scatter plot of CDK7 and pMED1 expression in tissue of (**A**) PTs, (**B**) LNs, (**C**) DMs, and (**D**) RDs. A trendline is shown. Irrespective of the tissue type CDK7 and pMED1 levels showed a significant correlation (DMs *p* = 0.0019, the rest *p* < 0.001), while the PCC ranged from 0.39 for LNs to 0.71 for DMs.

**Figure 6 cancers-14-00492-f006:**
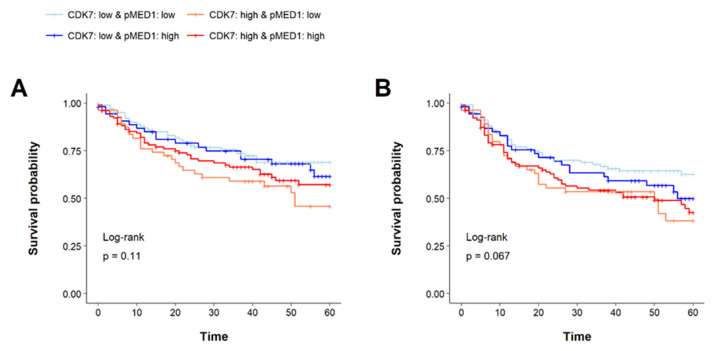
Kaplan–Meier graphs with *p*-values acquired from log-rank tests of (**A**) 5-year OS and (**B**) 5-year DFS. The cohort was stratified into four groups based on CDK7 and pMED1 expression. The expression above the median for the respective protein was considered as a high expression, the expression below the median as low expression. Based on this classification, four groups were formed to display all possible combinations of expression patterns. The log-rank tests revealed no significant difference in 5-year OS or 5-year DFS for the four groups. Nonetheless, we observed that curves grouped by CDK7 expression. There was a trend of shorter 5-year OS and 5-year DFS for both groups with high CDK7 expression compared to the two groups with low CDK7 expression. This effect seemed to be predominant and independent of the pMED1 expression level.

**Table 1 cancers-14-00492-t001:** Overview of clinical characteristics of CDK7 high and low groups.

Variable	CDK7 Expression		Total (*n* = 326)	*p* Value
	CDK7 Low (*n* = 161)	CDK7 High (*n* = 165)		
**Age**				0.355
Missing	0	2	2	
Mean (SD)	61.826 (10.126)	62.890 (10.527)	62.361 (10.328)	
**Sex**				0.882
Missing	1	0	1	
Female	36 (22.5%)	36 (21.8%)	72 (22.2%)	
Male	124 (77.5%)	129 (78.2%)	253 (77.8%)	
**Alcohol**				0.384
Missing	5	6	11	
No	91 (58.3%)	85 (53.5%)	176 (55.9%)	
Yes	65 (41.7%)	74 (46.5%)	139 (44.1%)	
**Nicotine**				0.854
Missing	11	6	17	
No	16 (10.7%)	18 (11.3%)	34 (11.0%)	
Yes	134 (89.3%)	141 (88.7%)	275 (89.0%)	
**Packyears**				0.608
Missing	11	6	17	
Mean (SD)	39.173 (26.669)	37.673 (24.659)	38.401 (25.624)	
**p16**				0.618
Negative	121 (75.2%)	120 (72.7%)	241 (73.9%)	
Positive	40 (24.8%)	45 (27.3%)	85 (26.1%)	
**Location**				0.716
Missing	5	4	9	
CUP	1 (0.6%)	0 (0.0%)	1 (0.3%)	
Hypopharynx	20 (12.8%)	24 (14.9%)	44 (13.9%)	
Larynx	47 (30.1%)	46 (28.6%)	93 (29.3%)	
Oral Cavity	33 (21.2%)	41 (25.5%)	74 (23.3%)	
Oropharynx p16-	28 (17.9%)	22 (13.7%)	50 (15.8%)	
Oropharynx p16+	27 (17.3%)	28 (17.4%)	55 (17.4%)	
**Recurrence**				0.431
No	123 (76.4%)	132 (80.0%)	255 (78.2%)	
Yes	38 (23.6%)	33 (20.0%)	71 (21.8%)	
**T-Stage**				0.816
Missing	2	0	2	
T (1,2)	83 (52.2%)	84 (50.9%)	167 (51.5%)	
T (3,4)	76 (47.8%)	81 (49.1%)	157 (48.5%)	
**N-Stage**				0.228
Missing	1	1	2	
N-	77 (48.1%)	68 (41.5%)	145 (44.8%)	
N+	83 (51.9%)	96 (58.5%)	179 (55.2%)	
**M-Stage**				0.112
Missing	0	1	1	
M-	145 (90.1%)	138 (84.1%)	283 (87.1%)	
M+	16 (9.9%)	26 (15.9%)	42 (12.9%)	
**UICC Stage**				0.566
Missing	1	0	1	
UICC (1,2)	60 (37.5%)	67 (40.6%)	127 (39.1%)	
UICC (3,4)	100 (62.5%)	98 (59.4%)	198 (60.9%)	
**Grading**				0.953
Missing	3	0	3	
G (1,2)	123 (77.8%)	128 (77.6%)	251 (77.7%)	
G (3,4)	35 (22.2%)	37 (22.4%)	72 (22.3%)	

**Table 2 cancers-14-00492-t002:** Cox regression analysis of CDK7 expression. * *p* < 0.05.

**5-Year Overall Survival**	**Univariable Survival Analysis**			**Multivariable Survival Analysis**		
**Variable**	**HR**	**95% CI**	***p* Value**	**HR**	**95% CI**	***p* Value**
Alcohol consumption	1.74	1.23–2.45	0.002	1.17	0.78–1.76	0.457
Packyears	1.01	1.00–1.02	<0.001	1.01	1.00–1.02	0.018 *
p16 status	0.48	0.31–0.74	<0.001	0.70	0.40–1.22	0.205
T-stage	2.29	1.61–3.25	<0.001	1.24	0.73–2.12	0.422
M-stage	1.98	1.31–2.99	<0.001	2.10	1.32–3.35	0.002 *
UICC	2.83	1.91–4.19	<0.001	1.87	1.00–3.50	0.050
CDK7	1.48	1.02–2.15	0.038	1.50	1.01–2.22	0.045 *
**5-Year Disease-Free Survival**	**Univariable Survival Analysis**			**Multivariable Survival Analysis**		
**Variable**	**HR**	**95% CI**	***p* Value**	**HR**	**95% CI**	***p* Value**
Age	1.02	1.00–1.03	0.012	1.03	1.01–1.05	0.007 *
Alcohol consumption	1.47	1.09–1.98	0.011	1.25	0.86–1.82	0.250
Packyears	1.01	1.00–1.01	0.022	1.00	1.00–1.01	0.453
p16 status	0.47	0.32–0.68	<0.001	0.48	0.29–0.81	0.006 *
T-stage	2.13	1.57–2.87	<0.001	1.44	0.88–2.36	0.144
M-stage	1.66	1.14–2.43	0.008	1.80	1.14–2.84	0.011 *
UICC	2.29	1.66–3.16	<0.001	1.40	0.80–2.43	0.237
CDK7	1.51	1.08–2.10	0.016	1.50	1.05–2.14	0.024 *

**Table 3 cancers-14-00492-t003:** Comparison of CDK7 expression regarding different clinicopathological parameters.

Clinicopathological Feature	*n*	*p* Value
Sex	Male (*n* = 72) vs. Female (*n* = 253)	0.59
Age	≤62 years (*n* = 152) vs. >62 years (*n* = 172)	0.17
Alcohol consumption	No (*n* = 176) vs. Yes (*n* = 139)	0.14
Nicotine consumption	No (*n* = 34) vs. Yes (*n* = 275)	0.4
Recurrent disease	No (*n* = 255) vs. Yes (*n* = 71)	0.75
p16 status	Negative (*n* = 241) vs. Positive (*n* = 85)	0.15
T-stage	T (1,2) (*n* = 167) vs. T (3,4) (*n* = 157)	0.73
N-stage	N0 (*n* = 145) vs. N+ (*n* = 179)	0.06
M-stage	M0 (*n* = 283) vs. M+ (*n* = 42)	0.06
UICC-stage	UICC (I,II) (*n* = 127) vs. UICC (III,IV) (*n* = 198)	0.89
Grading	G (1,2) (*n* = 251) vs. G (3,4) (*n* = 72)	0.77

## Data Availability

The data presented in this study are available on request from the corresponding author. The data are not publicly available due to restrictions in the privacy policy.

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
