# Peer review of "CDK7 Predicts Worse Outcome in Head and Neck Squamous-Cell Cancer"

_cancers, 2022, doi:10.3390/cancers14030492_

Round 1

Reviewer 1 Report

The authors report the association between CDK7 and pMED1 with prognosis in HNSCC. The paper is well written.

In the introduction, the authors should spend some time on negative and positive prognostic markers in HNSCC, notably p16 and CDKN2a (hypermethylation and underexpressed), as well as TP53 and why some of these markers are NOT used clinically right now.

In the methods section, some attention should be given to the validation and verification process used for IHC of CDK7 and pMED1 or refer to papers that specifically address this as this is not an IHC method is use clinically.

In the statistical section, the authors should review the use of p16 as a factor used in the multivariate analysis. p16 has proven pronostic value in oropharynx specifically (and unconfirmed in hypopharynx). Therefore, using it for the totality of patients is unsound. I would therefore ask the authors to conduct the analysis using p16 as a prognostic factor ONLY for OP patients. This might lead to differences in some of the associations reported and in the conclusions of the authors.

In the discussion, the authors should conclude by explaining or citing therapeutic avenues relevant to the demonstration made. In fact, the demonstration that CDK7 is possibly a prognostic factor does not preclude that it is an adequate therapeutic target. Moreover, considering the prognostic value, the authors should propose a validation cohort to support the results of this testing cohort before concluding on the proposal to use it clinically to establish a prognosis or a potential target of therapy.

Author Response

Point 1: In the introduction, the authors should spend some time on negative and positive prognostic markers in HNSCC, notably p16 and CDKN2a (hypermethylation and underexpressed), as well as TP53 and why some of these markers are NOT used clinically right now.

Response 1: Dear reviewer thank you for your points to improve the manuscript. We added a paragraph in the introduction section of the text.

“Besides, Hypermethylation of Cyclin-dependent kinase inhibitor 2A (CDKN2a), a gene coding for tumor suppressor genes p14ARF and p16INK4a, is involved in tumorigenesis of HNSCC and has also been investigated as a prognostic marker for this entity. Methylation transcriptionally silences genes and consequently decreases protein levels. However, investigation results varied between the studies regarding prognostic implications of CDKN2a (doi:10.1159/000494473.). Tumor suppressor gene p53 is one of the most commonly altered genes in HNSCC. Tumors harboring high-risk mutations showed worse outcomes (doi:10.1158/0008-5472.CAN-14-2735.). Nonetheless, an IHC study carried out to evaluate the association of p53 protein expression to clinicopathological features of an HNSCC cohort could not prove prognostic implications (doi:10.23937/2378-3419/1410122.).”

Point 2: In the methods section, some attention should be given to the validation and verification process used for IHC of CDK7 and pMED1 or refer to papers that specifically address this as this is not an IHC method is use clinically.

Response 2: Thank you for these thoughts. As a certified and accredited institute for pathology, we are specialized in establishing and performing IHC stains for clinical as well as research purposes. For the pMED1 and CDK7, we performed positive and negative control experiments and titrated the correct dilution by visual evaluation. Therefore, we added the following clause in the Materials/Methods section.

“For the staining of CDK7 and pMED1, the following antibodies were used at the indicated dilution after successful control tissue staining according to data sheets: CDK7 (mouse monoclonal, CDK7 (MO1) Mouse mAb, 1:500, Cell Signaling, Danvers, MA, USA) and pMED1 (rabbit polyclonal, Anti-TRAP220/MED1 (phospho T1457) antibody, 1:100, Abcam, Cambridge, UK).”

Point 3: In the statistical section, the authors should review the use of p16 as a factor used in the multivariate analysis. p16 has proven pronostic value in oropharynx specifically (and unconfirmed in hypopharynx). Therefore, using it for the totality of patients is unsound. I would therefore ask the authors to conduct the analysis using p16 as a prognostic factor ONLY for OP patients. This might lead to differences in some of the associations reported and in the conclusions of the authors.

Response 3: Thank you for this remark. We carefully considered only using p16 as a prognostic marker for the OPSCC subcohort. However, there is preliminary evidence that p16 also seems to play a prognostic role in non-oropharyngeal HNSCC, please see (doi: 10.1093/jnci/djy072.). Therefore, in previous publications concerning protein expression studies in our HNSCC, we regularly included p16 as a prognostic marker for the totality of the cohort, as it is significant in univariable cox regression (doi: 10.3390/ijms21155527., doi: 10.3390/ijms21155508., doi: 10.3390/ijms21030854.). We hope this sufficiently offers a rationale as to why we conducted the multivariable cox regression this way. However, we reanalyzed the OPSCC separately regarding p16 status. Here we could not verify CDK7 to be significantly prognostic, which is due to smaller sample size (n = 102, p = 0.1), but still high CDK7 expression was associated with worse 5-year overall survival. Nonetheless, as expected p16 indicated better 5-year overall survival in this subcohort (p = 0.0056).     

Point 4: In the discussion, the authors should conclude by explaining or citing therapeutic avenues relevant to the demonstration made. In fact, the demonstration that CDK7 is possibly a prognostic factor does not preclude that it is an adequate therapeutic target. Moreover, considering the prognostic value, the authors should propose a validation cohort to support the results of this testing cohort before concluding on the proposal to use it clinically to establish a prognosis or a potential target of therapy.

Response 4: This is a very good point. We revised the discussion to better put our findings into perspective and to call for experiments performed on an independent cohort by an independent research group. We also thought about splitting our cohort into a test and a validation set. However, we have refrained from this approach as it would have shrunken the sample size. We also added information on what our data MIGHT mean for future therapy. As our data does not support enough evidence to use CDK7 as a therapeutic target, we just speculate based on the current literature (doi: 10.1016/j.canlet.2019.11.027., doi: 10.1016/j.ccell.2019.11.003., doi: 10.1186/s13045-020-00926-x.) that a functional workup in HNSCC might elucidate therapeutic implications of CDK7. We hope you are content with the changes in the newly uploaded manuscript.

“We appreciate the limitations of our study as it provides correlation of clincal data and CDK7 expression determined via IHC staining. Further in-vitro studies are needed to investigate causation.”

“Finally, prospective studies are necessary to validate our findings on independent cohorts and to unravel molecular mechanisms by which CDK7 contributes to poor prognosis in HNSCC.”

Reviewer 2 Report

I have to congratulate the Authors for this interesting paper. It needs minor revisions of English text (lines 266, 314,315 as few examples)

Furthermore:

The role of CDK7 is well described and the data are clearly illustrated. it is surprising the high expression among oropharyngeal p16 positive patients, as we all know this population is characterized by a better prognosis while CDK7 over-expression indicates a more aggressive disease. I suggest the Authors to give an explanation of this unusual observation.

Similarly, I suggest to comment on CDK7 expression in infiltrate tumors, as these hot tumors, again, may have a better prognosis compared to the cold ones. Why this paradoxical expression?

And a comment on the potential effect of the checkpoint inhibitors combined with  the CDK7 inhibitors could better explain the relationship between these receptors.

Author Response

Point 1: I have to congratulate the Authors for this interesting paper. It needs minor revisions of English text (lines 266, 314,315 as few examples)

Response 1: Thank you! We employed a professional English spell check (Grammarly Inc.) to address language and grammar issues.

Point 2: Furthermore: The role of CDK7 is well described and the data are clearly illustrated. it is surprising the high expression among oropharyngeal p16 positive patients, as we all know this population is characterized by a better prognosis while CDK7 over-expression indicates a more aggressive disease. I suggest the Authors to give an explanation of this unusual observation.

Response 2: Dear reviewer, thank you for this hint. However, as we did not observe significant differences in the expression between different primary tumor locations, we are careful not to overinterpret this finding. For we can statistically not preclude the possibility that this result is purely caused by chance.    

Point 3: Similarly, I suggest to comment on CDK7 expression in infiltrate tumors, as these hot tumors, again, may have a better prognosis compared to the cold ones. Why this paradoxical expression?

Response 3: In previously published papers we found higher TPS in hot tumors (doi: 10.3389/fonc.2021.712788.) other studies regarding OSCC also found a positive correlation of PD-L1 expression and immune cell infiltration (doi: 10.3389/fimmu.2021.693881.). CDK7 was recently described to drive PD-L1 expression in an MYC-dependent manner (doi: 10.1186/s13045-020-00926-x.). We now speculate that CDK7 might drive PD-L1 expression in these tumors which might elicit immune infiltration. However since our study is the first to observe this linkage, prospective studies concerning the underlying mechanisms of the tumor microenvironment and CDK7 need to be conducted.    

Point 4: And a comment on the potential effect of the checkpoint inhibitors combined with  the CDK7 inhibitors could better explain the relationship between these receptors.

Response 4: Thank you for this comment to improve the manuscript. We added the paragraph according to your suggestion. Please see the Discussion section of the revised version for changes. There are mainly two mechanisms discussed. First CDK7 depletion leads to genomic instability increasing the mutational burden and eliciting immune cell infiltration (doi: 10.1016/j.ccell.2019.11.003.), second recently an MYC-dependent pathway has been discovered in NSCLC by which CDK7 increases PD-L1 expression (doi: 10.1186/s13045-020-00926-x.). The authors of the studies suggest that targeting CDK7 might synergistically boost immune cell infiltration and at the same time reduce PD-L1 mediated immune escape.

“We suppose that a combinational treatment might be of advantage for HNSCC patients as well, as immunotherapy develops as an integral component of therapeutic regimes. Especially patients with high CDK7 expression might benefit since CDK7 was recently discovered to increase PD-L1 expression through an MYC-dependent pathway in NSCLC. This mechanism leads to immune escape in the presence of infiltrating immune cells. The authors suggest that CDK7 inhibition sensitizes tumor cells to therapy with immune checkpoint inhibitors (doi:10.1186/s13045-020-00926-x.).”

Reviewer 3 Report

A very interesting original study investigating the potential role of CDK7 and pMED1 in head and neck squamous cell carcinoma, showing that CDK7 and pMED1 expression are correlated, and this correlation is not affected by tumor sites.  Patients with high CDK7 expression also reported a worse prognosis, suggesting CDK7 as a valuable new biomarker. The number of participants is adequate for such a study, and I was not able to find similar studies after quick research.Only minor queries before acceptance:

Conclusions should be expanded, better explaining future perspectives after this study's findings.

Page 2 line 54 you should add: "Although various other systemic and topical treatments have been proposed, none seems to improve patient's prognosis" and cite an article such as: "doi: 10.3390/medicina57060563. and doi: 10.3390/curroncol28040213."

Author Response

Point 1: Conclusions should be expanded, better explaining future perspectives after this study's findings.

Response 1: Dear reviewer thank you for taking the time to advance our manuscript. According to your recommendations, we added to the conclusion part of our paper.

“Our data suggest that CDK7 harbors valuable prognostic implications for HNSCC. Considering the poor prognosis and sparse landscape of suitable biomarkers and druggable targets in HNSCC it might be worth further pursuing CDK7 as a potential protein with implications for both prognosis and therapy, as there are already functioning inhibitors under investigation in the early stages of clinical trials (doi:10.1007/s10555-020-09885-8., doi:10.1158/1535-7163.MCT-16-0847.). Especially the linkage of CDK7 and immunotherapy might be of great interest for prospective research since checkpoint inhibition is becoming increasingly important in HNSCC treatment but mechanisms to boost therapy response remain poorly understood. Future studies may gather additional justification to include CDK7 in the routine pathological workup and potentially carry out clinical trials to assess inhibiting agents in HNSCC. Despite phosphorylation of MED1 by CDK7, their interactions seem to play a secondary role in the clinical context of HNSCC. Finally, prospective studies are necessary to validate our findings on independent cohorts and to unravel molecular mechanisms by which CDK7 contributes to poor prognosis in HNSCC.”

Point 2: Page 2 line 54 you should add: "Although various other systemic and topical treatments have been proposed, none seems to improve patient's prognosis" and cite an article such as: "doi: 10.3390/medicina57060563. and doi: 10.3390/curroncol28040213."

Response 2: Thank you for the referenced publication. We edited the text accordingly.

Reviewer 4 Report

In the present manuscript, the authors analyzed a large cohort of HNSCC patients with immunhistochemical staining of two prognostic candidates CDK7 and pMED1. As a result, upregulation of CDK7 was associated with a worse overall and disease free survival independent of other known prognostic factors indicating CDK7 as a novel therapeutic target for HNSCC patients.

However, the authors should carefully revise the references and imply them in the right order/position in the manuscript.

  • Line 86: Phosphorylation reinforces its association with Mediator complex [22], [25]

Reference 22 includes CDK7 and not MED1- probably a typo or editing problems?

  • Line 328: Ref 29 instead of 30?
  • Line 330: references seem not to fit to the statement / mentioned tumor entities

Further on, the authors stated in the „Introduction“ that for CDK7 and MED1 „ their prognostic role and potential interaction in HNSCC have not been studied yet.“ (line 94). The prognostic role of CDK7 was investigated (Ref. 22),  but not MED1 and their potential interaction in HNSCC have been studied in HNSCC.    

TPS is missing in the abbreviations

Author Response

Point 1: However, the authors should carefully revise the references and imply them in the right order/position in the manuscript.

Line 86: Phosphorylation reinforces its association with Mediator complex [22], [25]

Reference 22 includes CDK7 and not MED1- probably a typo or editing problems?

Line 328: Ref 29 instead of 30?

Line 330: references seem not to fit to the statement / mentioned tumor entities

Response 1: Dear reviewer thank you for your corrections and for checking the references. We also checked them before uploading however after formatting the document to plain text some references seemed to be switched. We now revised the references mentioned and checked all other references as well, and consequently updated the bibliography.

Point 2: Further on, the authors stated in the „Introduction“ that for CDK7 and MED1 „ their prognostic role and potential interaction in HNSCC have not been studied yet.“ (line 94). The prognostic role of CDK7 was investigated (Ref. 22),  but not MED1 and their potential interaction in HNSCC have been studied in HNSCC.    

Response 2: We appreciate your comment. CDK7 and pMED1 have yet not been studied using IHC on protein level in HNSCC and no study was published regarding their potential interaction in HNSCC. We, therefore, carried out our study. The study “Combinational therapeutic targeting of BRD4 and CDK7 synergistically induces anticancer effects in head and neck squamous cell carcinoma” by W. Zhang et al., performed an in-silico analysis using mRNA data from the TCGA to conclude prognostic implications of CDK7 (doi: 10.1016/j.canlet.2019.11.027.). 

Point 3: TPS is missing in the abbreviations

Response 3: We added TPS to the list of abbreviations.

Reviewer 5 Report

In the present manuscript the author found that high levels of CDK7 in patients tumor tissue point to shorter survival & shorter disease-free survival and overall claim that in HNSCC CDK7 might serve as a novel prognostic marker. The obtained results are interesting and may lead to finding a link between CDK7 and HNSCC pathogenesis. The manuscript is well-organized and well-written but lacks novelty and needs improvement. My comments are as follows:

  • It is interesting to discuss in more detail about the CDK7, as around a list of 790 markers are already reported across the different Cancer types and needs a strong statement as to why the author has only chosen the CDK7 for current study.
  • Despite intense research, reliable prognostic markers for oral cancer are still few and correlation of CDK7 with already known reliable markers (LIMA1, CALML5 and CD59) need to be mentioned.
  • If author is claiming about the CDK7 as novel prognostic marker, only immunohistochemical staining is not sufficient, author needs to provide additional data such as what is the status at transcriptional (RT-qPCR/RNA sequencing) and protein level (Western/Flow).
  • Lack the correlation between transcriptional and clinical data of the Cancer Genome Atlas (TCGA) database.
  • Figure1, quality is not good and hard (not confident) to understand the expression pattern and Figure 3, needs to be improved
  • Immunotherapy (pembrolizumab approved for HNSCC, 2019) is emerging as the fourth pillar of cure and nowhere the author discussed it, whether the patient will be detected with high CDK7 get benefited with immunotherapy or not.
  • Rather the mechanistic or experimental approach authors rely more on bioinformatics analysis but nowhere compare their findings with other databases or other published data.

Author Response

Point 1: It is interesting to discuss in more detail about the CDK7, as around a list of 790 markers are already reported across the different Cancer types and needs a strong statement as to why the author has only chosen the CDK7 for current study.

Response 1: Thank you for reviewing our manuscript. For this investigation, CDK7 was chosen as this protein was linked to worse prognosis and aggressive tumor phenotype throughout different entities, please see references (doi: 10.1016/j.ygyno.2019.11.004., doi: 10.1007/s10620-013-2597-x., doi: 10.1016/j.yexmp.2016.05.001., doi: 10.1016/j.pathol.2018.10.004., doi: 10.1158/1078-0432.CCR-15-1104.) as also cited in the introduction. Moreover, similar to ovarian cancer (doi: 10.1016/j.ygyno.2019.11.004.) HNSCC is subject to genomic heterogeneity and so far lacks well-defined targetable oncogenes. CDK7 as a transcriptional kinase is globally involved in the transcription of crucial genes in the transcriptional addiction of cancer cells (doi: 10.1016/j.canlet.2019.11.027.). Impairment of CDK7 might therefore exert beneficial effects in cancer therapy. This study now is exploratory and is the first insight into the potential linkage of CDK7 and patients’ clinical outcomes in general and shall provide a rationale for future mechanistic studies.     

Point 2: Despite intense research, reliable prognostic markers for oral cancer are still few and correlation of CDK7 with already known reliable markers (LIMA1, CALML5 and CD59) need to be mentioned.

Response 2: Dear reviewer thank you for taking the time to advance our manuscript and we are grateful for your advice. LIMA1, CALML5, and CD59 are suggested for prognostic implications by the protein atlas. However, as mentioned CDK7 was chosen for this investigation due to its role in various entities. A conducted TCGA in-silico analysis of CDK7 co-expression with the above mentioned markers revealed only very weak correlation (LIMA1: r = 0.15 p < 0.01, CALML5: r = 0.08, p = 0.127, CD59: r = 0.223, p > 0.001). Moreover, CDK7 is not involved in well-defined signaling pathways with these markers. Still, especially CALML5 has been validated in OTSCC (doi: 10.3390/cancers13102387.). However, in our study, we focused on HNSCC in general, as prior studies did (doi: 10.3390/ijms21155527., doi: 10.3390/ijms21155508., doi: 10.3390/ijms21030854., doi: 10.1158/1078-0432.CCR-20-0197., doi: 10.3389/fmed.2021.622330.). We are aware that site-specific analysis might be conducted but for the expected effect size we would call for larger site-specific cohorts.

Point 3: If author is claiming about the CDK7 as novel prognostic marker, only immunohistochemical staining is not sufficient, author needs to provide additional data such as what is the status at transcriptional (RT-qPCR/RNA sequencing) and protein level (Western/Flow).

Response 3: As pathologists we are interested in translational research and spatially determining protein abundance, hence for this investigation, IHC was the method of choice. We put our findings in perspective by saying CDK7 MIGHT be a POTENTIAL prognostic marker. Before establishing CDK7 in the clinical routine, validation on independent cohorts performed by independent research groups is a necessary step. But as stated, this study is the first of its kind looking at CDK7 in HNSCC. Follow-up projects are planned.     

Point 4: Lack the correlation between transcriptional and clinical data of the Cancer Genome Atlas (TCGA) database.

Response 4: Previous studies already linked CDK7 mRNA levels to survival data in the TCGA database and showed significantly better survival for the low expression group (doi: 10.1016/j.canlet.2019.11.027.). Still, mRNA levels are a poor measure to approximate protein expression (doi: 10.1038/nature22293.). Therefore, one might argue that protein expression and mRNA expression must be judged separately for prognostic implications.  

Point 5: Figure1, quality is not good and hard (not confident) to understand the expression pattern and Figure 3, needs to be improved.

Response 5: We uploaded all figures at the required resolution. Image sharpness might have been lost due to the word format. Expression patterns of proteins in Figure 1 are generally homogenous, meaning all cells within a core have about the same shade of staining, unlike markers like Ki-67 that show strong differentiation of negatively and positively stained cells/areas of proliferation within a sample. However, between samples, we could detect harsh differences in staining intensity. We hope this clarifies the mentioned staining patterns.  

Point 6: Immunotherapy (pembrolizumab approved for HNSCC, 2019) is emerging as the fourth pillar of cure and nowhere the author discussed it, whether the patient will be detected with high CDK7 get benefited with immunotherapy or not.

Response 6: We added the paragraph on the suspected effects of CDK7 expression on immunotherapy. Please see the Discussion section of the newly uploaded manuscript. However, we are cautious to conclude in this regard from our data as the cohort has been set up before the era of immunotherapy in HNSCC and because of the retrospective character of our investigation.

“We suppose that a combinational treatment might be of advantage for HNSCC patients as well, as immunotherapy develops as an integral component of therapeutic regimes. Especially patients with high CDK7 expression might benefit since CDK7 was recently discovered to increase PD-L1 expression through an MYC-dependent pathway in NSCLC. This mechanism leads to immune escape in the presence of infiltrating immune cells. The authors suggest that CDK7 inhibition sensitizes tumor cells to therapy with immune checkpoint inhibitors (doi:10.1186/s13045-020-00926-x.). We also could observe higher PD-L1 TPS in CDK7 overexpressing tumors, concluding that similar pathways might facilitate immune escape in HNSCC as well, and further supporting the statement that particularly patients with high CDK7 expression might profit from immunotherapy. Besides, we sought to investigate the association of CDK7 expression and immune infiltration for our cohort. As low CDK7, as mentioned above, might elicit immune infiltration. However, we found significantly higher CDK7 expression in immune infiltrated tumor tissue. Previously we described higher TPS in immune infiltrated tumors (doi:10.3389/fonc.2021.712788.). We suppose that CDK7 might drive higher PD-L1 expression in these tumors. However, since our study is the first to describe this linkage, further research is needed to characterize the tumor microenvironment according to CDK7 expression in HNSCC. We appreciate the limitations of our study as it provides correlation of clincal data and CDK7 expression determined via IHC staining. Further in-vitro studies are needed to investigate causation.“   

Point 7: Rather the mechanistic or experimental approach authors rely more on bioinformatics analysis but nowhere compare their findings with other databases or other published data.

Response 7: For this study, we performed IHC staining experiments to generate data and not rely on in silico research. Also, our study results point in the same direction as previously published IHC studies concerning other entities. Here we like to refer to the discussion of the present paper (see below). Generally, CDK7 has been associated with a worse prognosis and more aggressive phenotypes (doi: 10.1016/j.ygyno.2019.11.004., doi: 10.1007/s10620-013-2597-x., doi: 10.1016/j.yexmp.2016.05.001., doi: 10.1016/j.pathol.2018.10.004., doi: 10.1158/1078-0432.CCR-15-1104.). In HNSCC high mRNA expression was linked to worse survival in in-silico TCGA data (doi: 10.1016/j.canlet.2019.11.027.), and protein expression studies investigating OSCC a closely related carcinoma showed prognostic implications of CDK7 (doi: 10.1016/j.pathol.2018.10.004.).

“CDK7 overexpression has been observed in many cancer entities and generally indicates a worse prognosis and is associated with aggressive clinicopathological features. In oral squamous cell cancer, CDK7 overexpression was associated with shortened OS and DFS, as well as linked to higher T-stage (doi:10.1016/j.pathol.2018.10.004.). Similarly, studies conducted on gastric cancer, esophageal squamous cell carcinoma, and ovarian cancer showed worse survival, higher grading, and advanced tumor stages related to increased CDK7 levels (doi:10.1016/j.ygyno.2019.11.004., doi:10.1016/j.yexmp.2016.05.001., doi:10.1007/s10620-013-2597-x.).“